# Dietary Patterns Associated with Abnormal Glucose Tolerance following Gestational Diabetes Mellitus: The MyNutritype Study

**DOI:** 10.3390/nu15122819

**Published:** 2023-06-20

**Authors:** Farah Yasmin Hasbullah, Barakatun-Nisak Mohd Yusof, Sangeetha Shyam, Rohana Abdul Ghani, Hannah Izzati Mohamed Khir

**Affiliations:** 1Department of Dietetics, Faculty of Medicine and Health Sciences, Universiti Putra Malaysia, Serdang 43400, Selangor, Malaysia; farahyasmin90@gmail.com (F.Y.H.); hannahkhir@gmail.com (H.I.M.K.); 2Diabetes Research Unit, Faculty of Medicine and Health Sciences, Universiti Putra Malaysia, Serdang 43400, Selangor, Malaysia; 3Institute for Social Science Studies, Putra Infoport, Universiti Putra Malaysia, Serdang 43400, Selangor, Malaysia; 4Unitat de Nutrició Humana, Departament de Bioquímica i Biotecnologia, Universitat Rovira i Virgili, 43204 Reus, Spain; sangeetha.shyam@urv.cat; 5Institut d’Investigació Sanitària Pere Virgili (IISPV), Hospital Universitari Sant Joan de Reus, 43204 Reus, Spain; 6Centro de Investigación Biomédica en Red de Fisiopatología de la Obesidad y Nutrición (CIBEROBN), Instituto de Salud Carlos III (ISCIII), 28029 Madrid, Spain; 7Centre for Translational Research, IMU Institute for Research and Development (IRDI), International Medical University (IMU), Kuala Lumpur 57000, Malaysia; 8Department of Internal Medicine, Faculty of Medicine, Universiti Teknologi MARA, Sungai Buloh 47000, Selangor, Malaysia; rohana1773@uitm.edu.my; 9The Institute of Pathology, Laboratory and Forensic Medicine (I-PPerForM), Universiti Teknologi MARA, Sungai Buloh 47000, Selangor, Malaysia

**Keywords:** dietary patterns, abnormal glucose tolerance, gestational diabetes mellitus

## Abstract

Abnormal glucose tolerance (AGT), which includes type 2 diabetes and pre-diabetes, is highly prevalent in women post gestational diabetes mellitus (post-GDM). Dietary patterns have been associated with the risk of developing AGT in women post-GDM, but evidence in Asian populations is sparse. This study aimed to determine the association between a posteriori dietary patterns and AGT in women post-GDM. This cross-sectional study recruited 157 women post-GDM (mean age 34.8 years) from Seri Kembangan Health Clinic and Universiti Putra Malaysia. AGT was diagnosed according to the Malaysian Clinical Practice Guidelines using a 75 g 2 h oral glucose tolerance test or HbA1c. Food intake was assessed using the 2014 Malaysian Adult Nutrition Survey food frequency questionnaire. Five dietary patterns were derived using principal component analysis: ‘Unhealthy’, ‘Fish-eggs-fruits-vegetables’, ‘Cereals-confectionaries’, ‘Legumes-dairy’, and ‘Meat-sugar-sweetened-beverages’. After adjusting for sociodemographic characteristics and total energy intake, the ‘Cereals-confectionaries’ dietary pattern was significantly associated with AGT (adjusted odds ratio 1.536, *p* = 0.049). Targeted lifestyle modification, including dietary intervention, for women post-GDM is warranted to reduce their risk of AGT and its complications.

## 1. Introduction

Gestational diabetes mellitus (GDM), defined as diabetes first diagnosed in the second or third trimester of pregnancy that was not overt diabetes before gestation [1], has numerous short- and long-term implications for both mother and offspring [2]. GDM is an independent predictor of abnormal glucose tolerance (AGT) later in life [3], which includes both pre-diabetes and type 2 diabetes (T2D). Although most GDM cases disappear after delivery [4], hyperglycemia persists in 20–60% of women with the development of overt T2D within 5 years postpartum [5]. This may reveal chronic insulin resistance and β-cell dysfunction predating their pregnancies that deteriorated over time, ultimately leading to postpartum diabetes [2]. The cumulative incidence of T2D ranged between 2.6 and over 70% between 6 weeks and 28 years postpartum [5]. Meanwhile, in Asian women post-GDM, the prevalence of pre-diabetes ranged between 3.9 and 50.9% and T2D between 2.8 and 58% [3]. Both GDM and AGT also increase the risk of cardiovascular diseases (CVD) [6,7]. Therefore, identifying risk factors of AGT in women post-GDM is crucial to facilitate disease prevention strategies in this high-risk subgroup.

Lifestyle interventions, including diet, have been demonstrated to modify the risk of AGT in women post-GDM [8,9]. Dietary components that are investigated in this population have predominantly been studied using a reductionist approach and often revolve around the intake of single nutrients [10,11,12,13] or specific food groups [10,11,12,14,15]. However, the recent focus in nutritional epidemiology studies has shifted to a more holistic analysis of dietary patterns instead [16,17]. In reality, humans do not consume food or nutrients in isolation. Dietary patterns provide a more comprehensive description of the overall diet, which includes the frequency and quantity of the varying combinations of food and nutrients consumed [17]. Dietary patterns also take into consideration the synergistic interactions between nutrients and other compounds across food combinations [18]. Thus, analyzing food and nutrient consumption as dietary patterns may offer a better prediction of disease risk [17].

The more prominent method to assess dietary patterns in women post-GDM appeared to be a priori, and assessed the adherence to dietary quality indices or healthy dietary guidelines including the alternate Mediterranean diet (aMED), Dietary Approaches to Stop Hypertension (DASH), or alternate Healthy Eating Index (aHEI) [19,20]. Nonetheless, both of these studies used data from the Nurses’ Health Study II, which involved mostly Caucasian women [19,20], whose dietary quality, nutrient intakes, and food sources had been reported to be different compared to Asian women [21,22]. While a priori indices evaluate the healthiness of a diet based on existing dietary guidelines, they are restricted by existing knowledge of diet–disease relationships [17].

On the other hand, a posteriori approaches, such as principal component analysis (PCA), help to characterize the existing dietary habits in a population. PCA evaluates the overall habitual diet and the diversity in dietary intake of a population [23]. During analysis, the uncorrelated patterns represent the various dietary characteristics of the population and may be used to predict health outcomes; hence, the results may be more meaningful and significant compared to single-nutrient or single-food dietary analyses [23]. Furthermore, several major dietary patterns derived using this method have been replicated across different populations [23].

Malaysia is also in the midst of the Asian epidemic of T2D [24]. Malaysia is composed of diverse ethnic groups, primarily Malays, Chinese, and Indians [24]. The Chinese and Indian communities in Malaysia are descendants of their ancestors in China and India, respectively, who migrated to Malaysia between the early 19th and mid-20th century [25]. If India and China are projected to exceed the world’s global estimates of T2D, their respective counterparts in Malaysia will be more affected [24]. The interactions between different ethnic groups in Malaysia introduced varying degrees of genetic combinations and cultural influences [26] which possibly influenced their diet as well. The epidemiological transition brought upon by urbanization has also influenced Malaysia to undergo a rapid nutrition transition, indicated by the modernization and Westernization of traditional diets [27]. Hence, using PCA to analyse dietary patterns facilitates in depicting diverse dietary habits and planning individualized dietary strategies, specifically in multi-ethnic populations such as Malaysia. Therefore, this study aimed to determine the a posteriori dietary patterns associated with AGT in women post-GDM.

## 2. Materials and Methods

### 2.1. Study Design and Participants

This was a cross-sectional study involving women post-GDM from the MyNutritype cohort. The study protocol has been previously published [28]. In brief, the primary objective of the MyNutritype study was to identify the nutritype signatures (combination of dietary patterns and metabolomic profile) associated with AGT in women post-GDM. This analysis achieved the first objective of the MyNutritype study, which was to determine dietary patterns associated with AGT in women post-GDM. The study was conducted at Seri Kembangan Health Clinic and Universiti Putra Malaysia. Data were collected at Universiti Putra Malaysia from January to April 2021, then at Seri Kembangan Health Clinic from November 2021 to February 2022 in compliance with the nationwide Movement Control Order (MCO) regulations. Malaysia is an ethnically and religiously diverse country. Hence, data collection was halted during Ramadan fasting and around major festive periods (such as Eid al-Fitr, Christmas, Chinese New Year, and Deepavali) to prevent short-term but potentially significant changes in dietary patterns, as they may not accurately depict their actual habitual intake [29]. The study was approved by the Medical Research Ethics Committee of the Ministry of Health Malaysia (NMRR-19-3482-50546) and the Research Committee of Universiti Putra Malaysia (JKEUPM-2019-464).

The study recruited 157 women post-GDM, aged 18–49, who were currently 6 weeks to 15 years postpartum. Participants were considered to have previous GDM if the diagnosis was documented as per Malaysian Clinical Practice Guidelines [30]. We excluded pregnant women; those recently hospitalized within the last 6 weeks; those with chronic medical problems including type 1 or type 2 diabetes, cancer, renal or liver disorders. Participants gave their written informed consent at enrolment.

### 2.2. Dietary Assessment

Food intake was assessed using a semi-quantitative food frequency questionnaire (FFQ) adapted from the 2014 Malaysian Adult Nutrition Survey [31]. The FFQ contains 14 food groups with a total of 165 raw or cooked food and beverage items. The food groups are cereals and grains; fast food; meat and poultry; fish and seafood; eggs; legumes; milk and dairy products; fruits; vegetables; coffee, tea, and sugar-sweetened beverages; alcoholic drinks; confectionaries; bread spreads; and condiments. A trained dietitian asked the participants about the types of food and beverages consumed within the past month, as well as the frequency of intake and portion size of the food. The portion size of a food item was assigned according to the Food Portion Sizes of the Malaysian Foods Album [32]. The total amount of consumed food was calculated as grams/day for each food item based on the conversion factor for the FFQ that has been previously published [33].

Dietary intake was analysed for participants with a plausible reporting of energy intake, i.e., 500–3500 kcal/day for women [34]. Energy and macronutrient intakes were calculated using the Nutritionist Pro software (V5.1.0, Axxya Systems, Redmond, WA, USA). Nutrient profiles of foods were obtained from the Nutrient Composition of Malaysian Foods database [35] on the Nutritionist Pro software. If unavailable, data were obtained from the Singapore Food Composition database [36], followed by the U.S. Department of Agriculture database [37].

Dietary patterns were derived using principal component analysis (PCA). Food items that were not consumed by any of the respondents were excluded from PCA, including bacon and alcoholic beverages. Plain water was also excluded as it contains no calories and does not contribute significantly to nutrient intake. The food and beverage items were re-grouped into 12 food groups based on nutrient profile similarities and previous local data [27,38]. The new food groups were cereals and grains; fast food; meat and poultry; fish and seafood; eggs; legumes; milk and dairy products; fruits and vegetables; coffee, tea, and sugar-sweetened beverages; confectionaries; sugar, honey, bread spreads, and creamer; and condiments. Data suitability was performed before performing PCA.

The Kaiser–Meyer–Olkin (KMO) value was 0.653 and Bartlett’s Test of Sphericity was *p* < 0.001, indicating that the sample size was suitable to perform PCA [39]. Factor scores were orthogonally rotated by Varimax transformation to enhance interpretability and loading differences [40]. The number of dietary patterns extracted was determined by eigenvalue >1.0 and scree plot (the number of data points above the inflection point). Factor loading scores of ≥ |0.3| indicated a high or low intake of that food group [40,41]. Negative factor loadings were taken to indicate foods that were excluded and positive factor loadings to indicate foods that were included within a specific dietary pattern. Correlations between PCA loading scores and nutrients were also performed in order to better characterize the dietary patterns derived from PCA. 

### 2.3. Diagnosis of Abnormal Glucose Tolerance

After an overnight fast of 8–12 h, all participants underwent a 75 g 2 h oral glucose tolerance test (OGTT) and had blood samples taken for HbA1c assessment. Based on the Malaysian Clinical Practice Guidelines [42], AGT was diagnosed if they had fasting plasma glucose ≥ 6.1 mmol/L, 2 h plasma glucose ≥ 7.8 mmol/L, or HbA1c ≥ 5.7%. Participants were then categorized into either normal glucose tolerance (NGT) or AGT group.

### 2.4. Other Variables

The study collected sociodemographic data on participants’ age, ethnicity, marital status, education level, working status, household income, and household size using a structured questionnaire. For working status, those who were housewives or students were considered as not working, whereas employed or self-employed participants were categorized as working. Monthly household income was classified into 3 groups: low- (<RM 4850), middle- (RM 4850–10,959), or high-income (>RM 10,959) [43]; USD 1 equals approximately MYR 4.30.

Obstetric history was obtained from participants’ antenatal records, such as a family history of diabetes, gravidity, parity, and GDM recurrence. Index GDM was defined as the most recent GDM pregnancy. Information on their index GDM included duration since index GDM diagnosis, pre-pregnancy BMI, gestational age during GDM diagnosis, delivery method, gestational weight gain, treatment, breastfeeding practices, infant birth weight, and postpartum weight retention. Pre-pregnancy BMI was classified based on the BMI cut-offs for Asian populations [44]. Macrosomia in the offspring of participants was defined as infant birth weight > 4.0 kg [30].

Participant’s height was measured using a stadiometer (SECA model 206, Vogel & Halke GmbH & Co., Hamburg, Germany). Weight and body fat percentage were measured using a body composition digital scale (Tanita Health Equipment Ltd., Tokyo, Japan). BMI was then calculated and categorized according to the BMI cut-offs for Asian populations [44]. Waist and hip circumferences were measured using a measuring tape (SECA model 203, Vogel & Halke GmbH & Co., Hamburg, Germany). Abdominal obesity was indicated by waist circumference ≥ 80 cm or a waist-to-hip ratio of > 0.8 [44,45]. Blood pressure was measured using a blood pressure monitor (OMRON Corporation, Kyoto, Japan) and classified according to the Adult Treatment Panel (ATP III) guidelines [46]. Apart from OGTT and HbA1c, participants’ fasting venous samples were also taken for insulin level and lipid profile. The standard procedures for anthropometric, biochemical, and clinical measurements above have been elaborated in detail [28].

Meanwhile, physical activity level was assessed using the International Physical Activity Questionnaire-Short Form (IPAQ-SF) [47]. The 7-item IPAQ-SF asked about participants’ frequency and duration of vigorous-intensity activities, moderate-intensity activities, walking, and sitting in the last 7 days. Total scores were calculated as the sum of the frequency (days) and duration (minutes) of vigorous-intensity, moderate-intensity, and walking activities, which were expressed in MET-minutes/week. Data on sitting were not included in the total score and were only reported in terms of median and interquartile range [48].

### 2.5. Statistical Analysis

Statistical tests were performed in SPSS software version 25.0, with statistical significance set at *p* < 0.05. Continuous data were reported in mean ± standard deviation (SD), and categorical data in numbers and percentages. Independent *t*-test or Pearson’s chi-squared tests were used to compare the characteristics between NGT and AGT groups. Binary logistic regression was used to determine the association of dietary patterns (exposure) with AGT (outcome). Dietary patterns were used as a continuous variable from the PCA derived scores. AGT prevalence was a dichotomous variable (yes or no). Other variables that were associated with AGT were entered into the regression model along with each dietary pattern. These were identified either from the literature or those that were significant at the bivariate level (*p* < 0.2), which included age, education, occupation, household size, family history of diabetes, and TEI. Odds ratios (ORs) with 95% confidence intervals (CIs) were reported for each regression model.

## 3. Results

A total of 57 (36.3%) participants were diagnosed with AGT. From this, 47 (29.9%) had pre-diabetes and 10 (6.4%) participants had T2D. Participants with AGT had significantly larger household sizes (*p* = 0.005) and had higher gravidity (*p* = 0.037), parity (*p* = 0.004), pre-pregnancy BMI (*p* = 0.001), current BMI (*p* = 0.007), waist circumference (*p* = 0.012), and hip circumference (*p* = 0.040). They were also more likely to have had recurrent GDM (*p* = 0.023). Regarding their biochemical profile, the AGT participants had significantly higher levels of fasting insulin (*p* < 0.001), total cholesterol (*p* = 0.024), triglyceride (*p* = 0.015), LDL cholesterol (*p* = 0.010), non-HDL cholesterol (*p* = 0.001), and a higher ratio of total to HDL cholesterol (*p* < 0.001); but they had lower HDL cholesterol levels (*p* = 0.021). Other variables were comparable between the two groups (Table 1).

Dietary intake was only analysed in 150 (95.5%) participants with plausible dietary reporting. From this, 97 (97.0%) participants were from the NGT, and 53 (93.0%) were from the AGT group. Total fat as a proportion of total energy intake (% TEI) and saturated fat intake were significantly lower in the AGT group (*p* = 0.033 and 0.043, respectively). In terms of food groups, the AGT participants consumed a significantly lower amount of eggs (*p* = 0.034) and milk and dairy products (*p* = 0.012) compared to the NGT participants. Other dietary variables were not different between the two groups (Table 2).

Based on eigenvalues > 1.0 and scree plotting during PCA, five dietary patterns emerged, which explained 59.8% of the total variance (Appendix A). The first dietary pattern, ‘Unhealthy’, was characterized by high intakes of fast food; milk and dairy products; coffee, tea, and sugar-sweetened beverages (SSB); sugar, honey, bread spreads, and creamer; and condiments. This dietary pattern contributed to 19.9% of the total variance. This dietary pattern was labelled ‘Unhealthy’ because most of the foods that loaded highly in this pattern were demonstrated to exert detrimental effects on metabolic health. Although total dairy, low-fat dairy, cheese, and yogurt intakes have been shown to have protective effects against T2D [49], this food group also included both full-fat and low-fat milk/dairy, and fresh milk and flavoured milk (e.g., chocolate milk). Flavoured milk is often sweetened; one cup of chocolate milk contains approximately 16g of total sugars, compared to full-fat milk which has 11 g of total sugars [35]. Fast food has been associated with weight gain and insulin resistance, which could lead to an increased risk of obesity and T2D [50]. Coffee and tea were mostly consumed with condensed milk or sugar among Malaysian adults, and SSB has been associated with increased cardiometabolic risks in the Malaysian population [51]. Sugar, honey, bread spreads, and creamer (this food group also contained condensed milk) are high in added sugar, which has been shown to increase cardiovascular risk by increasing LDL cholesterol and triglyceride levels [52]. Finally, condiments, which include table salt, a variety of sauces (soy, chili, ketchup, oyster, fish sauces), and dipping sauces (*sambal*, *budu*, *cincalok*) are high in sodium content, which increases blood pressure levels [53].

The second dietary pattern derived, ‘Fish-eggs-fruits-vegetables’, included high intakes of fish and seafood; eggs; and fruits and vegetables; and explained 12.5% of the total variance. Marine fish and green leafy vegetables are among the top ten foods most consumed by Malaysian adults [54]. Aside from eggs, the foods in this dietary pattern were included in the ‘Mainly healthy’ dietary pattern, which was found to be inversely correlated with the risk of T2D [55]. While this dietary pattern among our subjects included several healthy food groups and was correlated with higher fiber intake, it also showed the highest correlation among the groups with sugar intake (Table 3). The correlation with energy intake also suggests that attention to portion control may sometimes be required in those adhering to this dietary pattern.

The third dietary pattern, ‘Cereals-confectionaries’ consisted of high intakes of cereals and grains and confectionaries, but low intake of milk and dairy products (accounting for 9.5% of the total variance). The cereals included a variety of grains, cereals, and cereal products, including rice (white rice, brown rice, flavoured rice), noodles, pasta, corn, breakfast cereals, oats, breads, and local flatbread. Confectionaries consisted of sweet or savoury foods and snacks, such as local *kuih*, biscuits, chocolate, candies, and pastries.

Next, the ‘Legumes-dairy’ dietary pattern had high intakes of legumes, and milk and dairy products (explaining 9.2% of the total variance). Total legume intakes were associated with an inverse risk of T2D [56], as well as dairy intake [49]. As in dietary pattern 1 (‘Unhealthy’), however, the milk and dairy products in this dietary pattern included both low-fat and full-fat milk/dairy, and fresh milk and flavoured milk. Thus, this dietary pattern could have a mixed effect on the risk of AGT.

The fifth dietary pattern, ‘Meat-sugar-sweetened beverages’, which accounted for 8.7% of the total variance, was characterized by high intakes of meat and poultry and SSB. The meat in this dietary pattern referred to meat and poultry (including beef, mutton, chicken, duck, pork) and their products (such as internal organs, ham, bacon, luncheon meat, meatballs). Red and processed meat have been associated with a positive risk of T2D, whereas poultry has been with decreased risk of T2D [55]. SSB included coffee and tea, which are usually mixed with sugars among Malaysians [51], as well as non-alcoholic sugary drinks such as malted drinks, yogurt drinks, carbonated drinks, and fruit juices (Figure 1).

Thus, each dietary pattern identified potential targets to improve the healthfulness of the diet, facilitating the individualization of recommendations in clinical practice.

Correlations of dietary patterns with energy and macronutrient intakes are shown in Table 3. The ‘Fish-eggs-fruits-vegetables’ dietary pattern had a moderate and positive correlation with protein and fiber intakes (both *p* < 0.001). The ‘Cereals-confectionaries’ dietary pattern was moderately positively correlated with carbohydrate intake (*p* < 0.001). Meanwhile, the ‘Meat-sugar-sweetened beverages’ dietary pattern had a moderate, positive correlation with intakes of protein, total fat, and saturated fat (all *p* < 0.001) (Table 3).

After adjusting for age, education level, occupation, household size, family history of diabetes, and TEI, the ‘Cereals-confectionaries’ dietary pattern was significantly associated with increased odds of AGT (adjusted odds ratio (AOR) 1.536, *p* = 0.049). The model contributed 15.3% to the total variance. Other dietary patterns did not have significant associations with AGT (Table 4).

The top five food sources of the ‘Cereals-confectionaries’ dietary pattern were white rice (58.9%), white bread (7.8%), brown rice (5.2%), flavoured rice (4.5%), and local *kuih* (3.2%) (Figure 2). Flavoured rice is rice cooked in a variety of dishes; examples are fried rice, Hainanese chicken rice, and *nasi lemak* (rice cooked in coconut milk, accompanied with condiments that include anchovy *sambal*, boiled egg, cucumber, fried anchovies, and ground nuts). *Kuih* is a wide range of local delicacies, which can be sweet or savoury; examples include curry puffs, *kuih lapis* (layered cake made from rice flour), and *kuih seri muka* (pandan-layered cake made from glutinous rice and coconut milk).

## 4. Discussion

The prevalence of AGT in this study was 36.3%, which aligned with the prevalence of AGT in Asian women post-GDM [3]. Women post-GDM who had AGT had a larger household size and higher parity and gravidity, rate of GDM recurrence, pre-pregnancy BMI, current BMI, waist and hip circumferences, and fasting insulin level. The AGT group also had higher levels of total cholesterol, triglycerides, LDL cholesterol, non-HDL cholesterol, total to HDL cholesterol ratio, and lower HDL cholesterol level, demonstrating their early pre-disposition to CVD.

After adjustment for covariates, the ‘Cereals-confectionaries’ dietary pattern was significantly associated with increased odds of AGT. This dietary pattern was positively and significantly associated with total carbohydrate intake. The main food sources that contributed to this dietary pattern were, in order of weightage: white rice, white bread, brown rice, flavoured rice, and local *kuih*. White rice is a daily staple in Malaysia; about 89.8% of Malaysians consumed white rice twice daily (approximately 2.5 plates or 250 g/day) [54]. Findings from several Asian countries reported a higher incidence of T2D for those consuming the highest amount of white rice (pooled risk ratio, RR 1.25, 95% confidence interval (CI) 1.17–1.33), and the risk was higher in women (RR 1.58, 95% CI 1.26–1.99) [57]. Additionally, there appeared to be a linear dose–response relationship for each 50 g/day of white rice or whole grains consumed (RR 1.000316, 95% CI 1.00039–1.000465). In contrast, each 50 g/day increment in brown rice intake reduced the risk of T2D by 13% (RR 0.87, 95% CI 0.80–0.94) [58]. Nonetheless, white rice, white bread, brown rice, flavoured rice, and local *kuih* tend to have medium or high glycemic index values. White rice, white bread, and local *kuih* (such as spring rolls), especially, have high glycemic index (GI) values of 90, 82, and 78, respectively [59]. Diets high in GI or glycemic load (GL) have been hypothesized to elevate postprandial glucose and insulin responses, leading to insulin resistance and hyperglycemia [60,61].

In a multi-ethnic, diverse population undergoing rapid nutrition transitions like Malaysia, it is specifically useful to make the effort to identify existing variations in dietary intakes. Unlike the one or two major dietary patterns that are reported in Western countries (such as ‘Healthy/Prudent’ and ‘Unhealthy’ dietary patterns), the diversity in Asian countries, including Malaysia, informs researchers on the practical use of culturally tailored dietary patterns, as the ‘One size fits all’ approach may not be suitable. The PCA-derived dietary patterns in this study did not mimic the usual dietary patterns reported in Western countries due to differences in dietary habits. For instance, the ‘Sugar-sweetened-beverages (SSB)’ dietary pattern was associated with increased cardiometabolic risks in Malaysian multi-ethnic healthy adults [51]. Additionally, a dietary pattern characterized by a higher intake of dim sum, meat/processed meat, SSB and sweetened foods, and fried foods was associated with a higher risk of T2D in Singaporean Chinese adults [62]. These results were almost similar to our findings on women post-GDM, and informed the need for more tailored dietary intervention strategies to facilitate disease prevention. Dietary patterns identified in this study indicated the most common foods in the diets of women post-GDM.

In addition to having a high carbohydrate intake, the AGT group had lower fat (as a proportion of TEI) and saturated fat intakes. Current evidence indicates that the type of fat consumed, and not the total amount, could modify the risk of chronic diseases including T2D [63,64]. Moreover, practicing high-carbohydrate, low-fat diets, especially without considering the quality of fat, may lead to excess consumption of refined carbohydrates, increasing the risk of cardiometabolic diseases [18]. This shows another benefit of studying dietary patterns over single-nutrient intakes, as we can observe the quality of the dietary patterns derived from food groups. In addition to being a high-GI dietary pattern, biscuits and local *kuih* are also among the most consumed food items in the country [33,54]. These traditional delicacies are commonly made with white sugar, wheat flour, glutinous rice, and/or coconut milk, and tend to be deep-fried. Thus, the ‘Cereals-confectionaries’ dietary pattern can be considered as being low in both carbohydrate and fat quality. This is demonstrated by the AGT group in our study, who still had a suboptimal lipid profile despite a lower fat intake.

The mechanism linking carbohydrate-rich dietary patterns and AGT has been explored in exposome research. The consumption of refined wheat bread was associated with elevated postprandial levels of branched-chain amino acids (BCAAs) in a crossover meal study involving postmenopausal women [65]. Elevated BCAA levels were also associated with ‘bread, margarine, and processed meat’ and ‘potatoes, cornflakes, dairy products, raw vegetables, and desserts’ dietary patterns in two different European Prospective Investigation into Cancer and Nutrition (EPIC)-Potsdam cohorts [66,67]. BCAAs, which include isoleucine, leucine, and valine were reported to be high in individuals with obesity [68] and are biomarkers of insulin resistance and future risk of diabetes [69]. Leucine-mediated stimulation of the mammalian target of rapamycin complex 1 (mTORC1), leading to early decoupling of insulin signalling and the onset of insulin resistance, is thought to be the link between elevated BCAAs and diabetes [68]. Additionally, a dysregulated BCAA metabolism was also hypothesized to lead to the accumulation of mitotoxic metabolites, promoting diabetes through β-cell mitochondrial dysfunction, stress signalling, and apoptosis [68].

This study has a few limitations. The cross-sectional design of the study did not allow cause and effect to be established; only the correlation between dietary patterns and AGT at a single time point could be determined. Secondly, the study was conducted in Malaysia, and food intake was assessed using a culturally specific FFQ that is unique to this country. Thus, the results may not be generalizable to other countries.

Moreover, the analysis of a posteriori dietary patterns using the PCA method may be subject to variation or researcher bias [70]. Decisions that could influence the validity of the extracted components include the method of rotation used, the threshold level of eigenvalues, the type of transformation, the number of factors to extract, and the labelling of food groups [17,71]. The subjective decision making during PCA, and the heterogeneity of the population, has made comparisons with other studies difficult. However, the study maintained primary considerations when conducting PCA, such as the nutrient composition of the food [72], the key foods constituting dietary patterns that are relevant in preventing AGT [72], and the rapidly shifting dietary landscape in Malaysia (including the Westernization and modernization of traditional diets and the availability of convenience or processed foods) [27,38]. Nonetheless, the dietary patterns derived using the PCA approach here explained 59% of the population variance. This compares more favourably to previous explorations in Malaysia [27,38].

Additionally, although the FFQ is a useful, common instrument in nutrition epidemiology research, it can be subject to recall errors, nutrient estimation errors, and other biases [73]. Therefore, further research should include the objective assessment of dietary patterns to improve its overall accuracy, such as by using high-throughput metabolomic analysis. These metabolite biomarkers of dietary patterns, measured in plasma, serum, or urinary samples, can objectively assess diet and validate the use of subjective dietary assessment methods including FFQs [73]. Finally, despite adjusting for several key confounders, the presence of residual confounding cannot be discounted. Nevertheless, the directionality of the results and the magnitude of the associations remain stable in the different models.

## 5. Conclusions

The study identified five dietary patterns derived from a culturally unique FFQ that reflects the typical Asian diet in women post-GDM. The consumption of ‘Cereals-confectionaries’ dietary patterns, which correlated positively with total carbohydrate intake, was significantly associated with increased odds of AGT in women post-GDM. Women post-GDM who developed AGT also had higher gravidity and parity, BMI, waist and hip circumferences, fasting insulin level; suboptimal lipid profile; and were likely to be diagnosed with GDM more than once. Findings from this study help to identify the characteristics of women post-GDM who may be targeted for dietary intervention to help reduce their risk of developing AGT and its complications. Further research should include the objective assessment of diet, such as through using a metabolomics approach, to improve the overall accuracy of dietary pattern assessment.

## Figures and Tables

**Figure 1 nutrients-15-02819-f001:**
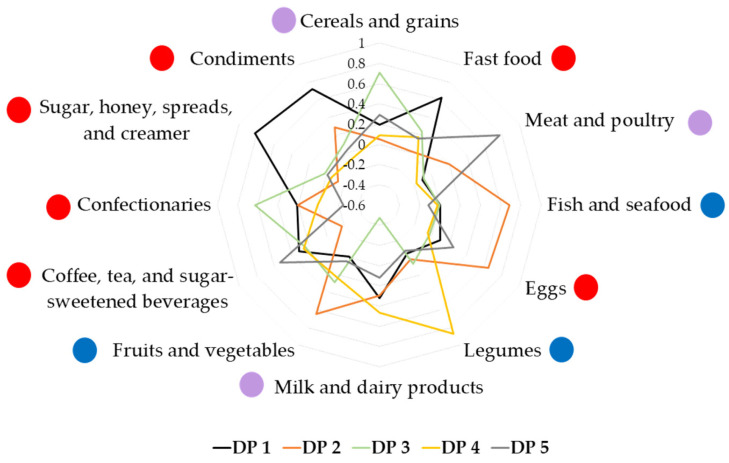
Factor loading scores of dietary patterns identified using principal component analysis. DP 1—‘Unhealthy’ dietary pattern; DP 2—‘Fish-eggs-fruits-vegetables’ dietary pattern; DP 3—‘Cereals-confectionaries’ dietary pattern; DP 4—‘Legumes-dairy’ dietary pattern; DP 5—‘Meat-sugar-sweetened beverages’ dietary pattern. Red circle 

: food group perceived to be healthy; blue circle 

: food group perceived to be unhealthy; purple circle 

: food group consisting of healthy and unhealthy food items.

**Figure 2 nutrients-15-02819-f002:**
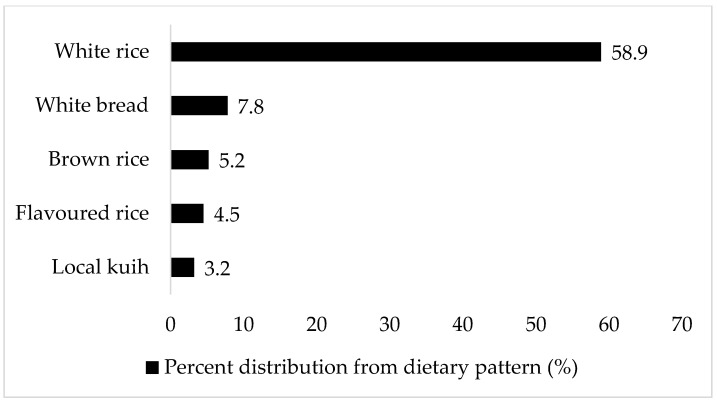
Top food sources of ‘Cereals-confectionaries’ dietary pattern. The food item’s percent distribution from the dietary pattern is based on the total amount consumed (g/day).

**Table 1 nutrients-15-02819-t001:** Comparison of characteristics between normal glucose tolerance and abnormal glucose tolerance groups (*n* = 157).

Variables	NGT (*n* = 100)	AGT (*n* = 57)	*p*-Value
	Mean ± SD or *n* (%)		
Sociodemographic background			
Age (years)	34.3 ± 5.9	35.6 ± 5.1	0.188
Age distribution			
18–24	1 (1.0)	2 (3.5)	0.134
25–34	56 (56.0)	22 (38.6)	
35–44	38 (38.0)	30 (52.6)	
45–49	5 (5.0)	3 (5.3)	
Ethnicity			
Malay	82 (82.0)	49 (86.0)	
Chinese	11 (11.0)	2 (3.5)	0.225 ^a^
Indian	4 (4.0)	5 (8.8)	
Others	3 (3.0)	1 (1.8)	
Marital status			
Married	99 (99.0)	56 (98.2)	0.999 ^a^
Divorced/widowed	1 (1.0)	1 (1.8)	
Education level			
Primary education	1 (1.0)	0 (0.0)	
Secondary education	42 (42.0)	32 (56.1)	0.151 ^a^
Tertiary education	57 (57.0)	25 (43.9)	
Working status			
Not working (housewife/student)	27 (27.0)	23 (40.4)	0.084
Working (employed/self-employed)	73 (73.0)	34 (59.6)	
Monthly household income (RM)	6300 ± 3889	6211 ± 4568	0.898
Low-income group (<RM4850)	35 (35.0)	26 (45.6)	
Middle-income group (RM4850–10,959)	56 (56.0)	25 (43.9)	0.327
High-income group (>RM10,959)	9 (9.0)	6 (10.5)	
Household size	4 ± 1	5 ± 1	0.005 *
General obstetric history			
Family history of diabetes	68 (68.0)	45 (78.9)	0.142
Gravidity	3 ± 2	4 ± 2	0.037 *
Parity	2 ± 1	3 ± 1	0.004 *
GDM recurrence	16 (16.0)	18 (31.6)	0.023 *
Obstetric history during index GDM			
Duration since index GDM (years)	2.6 ± 3.6	2.4 ± 3.4	0.810
Pre-pregnancy BMI (kg/m^2^)	23.6 ± 4.0	26.3 ± 5.0	0.001 *
BMI categories			
Underweight (<18.5 kg/m^2^)	10 (10.0)	2 (3.5)	
Normal (18.5–22.9 kg/m^2^)	32 (32.0)	12 (21.1)	0.010 *
Overweight (23.0–24.9 kg/m^2^)	27 (27.0)	10 (17.5)	
Obese (≥25.0 kg/m^2^)	31 (31.0)	33 (57.9)	
Gestational weight gain (kg)	10.8 ± 7.0	9.9 ± 5.5	0.413
Postpartum weight retention (kg)	5.1 ± 6.8	4.1 ± 6.0	0.365
Gestational age during diagnosis (weeks)	20.3 ± 7.7	18.6 ± 7.3	0.189
Delivery method			
Spontaneous vaginal delivery	66 (66.0)	35 (61.4)	0.454
Caesarean section	32 (32.0)	22 (38.6)	
Treatment			
Diet control only	87 (87.0)	47 (82.5)	0.439
Diet control with metformin/insulin	13 (13.0)	10 (17.5)	
Breastfeeding status			
Never breastfed	0 (0.0)	1 (1.8)	
Stopped breastfeeding	34 (34.0)	26 (45.6)	0.119 ^a^
Still breastfeeding	64 (64.0)	30 (52.6)	
Infant birth weight (kg)	3.05 ± 0.51	3.14 ± 0.52	0.313
Presence of macrosomia (>4.0 kg)	3 (3.0)	4 (7.0)	0.424 ^a^
Anthropometric and clinical measurements			
Height (m)	1.57 ± 0.06	1.56 ± 0.05	0.815
Current weight (kg)	63.3 ± 13.4	68.7 ± 13.5	0.016 *
Current BMI (kg/m^2^)	25.7 ± 5.0	27.9 ± 4.7	0.007 *
BMI categorization			0.014 *
Underweight (<18.5 kg/m^2^)	7 (7.0)	1 (1.8)	
Normal (18.5–22.9 kg/m^2^)	21 (21.0)	8 (14.0)	
Overweight (23.0–24.9 kg/m^2^)	24 (24.0)	6 (10.5)	
Obese (≥25.0 kg/m^2^)	48 (48.0)	42 (73.7)	
Waist circumference (cm)	85.5 ± 10.1	89.9 ± 11.5	0.012 *
Within recommendation (<80 cm)	30 (30.0)	11 (19.3)	
Abdominal obesity (≥80 cm)	70 (70.0)	46 (80.7)	0.142
Hip circumference (cm)	104.7 ± 10.3	108.2 ± 9.8	0.040 *
Waist-to-hip ratio	0.83 ± 0.14	0.83 ± 0.06	0.935
Within recommendation (≤0.8)	40 (40.0)	18 (31.6)	
Abdominal obesity (>0.8)	60 (60.0)	39 (68.4)	0.293
Systolic blood pressure (mmHg)	111 ± 15	112 ± 15	0.706
Diastolic blood pressure (mmHg)	79 ± 11	81 ± 9	0.422
Blood pressure category			
Normal	52 (52.0)	27 (47.4)	
Pre-hypertension	6 (6.0)	1 (1.8)	0.532 ^a^
Stage 1 hypertension	26 (26.0)	19 (33.3)	
Stage 2 hypertension	16 (16.0)	10 (17.5)	
Biochemical profile			
Fasting plasma glucose (mmol/L)	4.55 ± 0.43	5.32 ± 1.71	<0.001 *
2-h plasma glucose (mmol/L)	5.51 ± 1.18	8.15 ± 3.18	<0.001 *
HbA1c (%)	5.3 ± 0.2	6.0 ± 1.2	<0.001 *
Fasting insulin (uIU/mL)	6.44 ± 4.48	10.65 ± 9.01	<0.001 *
Total cholesterol (mmol/L)	5.14 ± 0.81	5.50 ± 1.01	0.024 *
Triglycerides (mmol/L)	1.03 ± 0.57	1.32 ± 0.76	0.015 *
HDL cholesterol (mmol/L)	1.59 ± 0.43	1.44 ± 0.36	0.021 *
LDL cholesterol (mmol/L)	3.07 ± 0.74	3.46 ± 0.96	0.010 *
Non-HDL cholesterol (mmol/L)	3.55 ± 0.80	4.07 ± 0.98	0.001 *
Total to HDL cholesterol ratio	3.4 ± 0.9	4.0 ± 1.0	<0.001 *
Physical activity			
Total physical activity level (MET-min/week)	2088 ± 401	1951 ± 387	0.822
Vigorous-intensity (MET-min/week)	482 ± 146	368 ± 98	0.584
Moderate-intensity (MET-min/week)	822 ± 183	768 ± 168	0.844
Walking (MET-min/week)	783 ± 188	814 ± 308	0.928
Sitting duration (minutes/day)	372 ± 20	363 ± 26	0.795

AGT: abnormal glucose tolerance; BMI: body mass index; NGT: normal glucose tolerance; RM: Ringgit Malaysia (the currency of Malaysia; USD 1 = ±MYR 4.30). Continuous data expressed in mean ± SD, analysed using independent *t*-test. Categorical data expressed in number (%), were analysed using Pearson’s chi-squared tests or ^a^ Fisher’s exact tests. * *p* < 0.05.

**Table 2 nutrients-15-02819-t002:** Comparison of intakes of energy, macronutrients, and food groups between normal glucose tolerance and abnormal glucose tolerance groups (*n* = 150).

Dietary Variables	NGT (*n* = 97)	AGT (*n* = 53)	*p*-Value
Energy and nutrient intakes			
Total energy intake (TEI) (kcal/day)	1738 ± 682	1681 ± 695	0.631
Carbohydrate intake			
Total amount (g/day)	232 ± 99	235 ± 104	0.856
As part of TEI (%)	54 ± 9	56 ± 8	0.113
Protein intake			
Total amount (g/day)	83 ± 40	80 ± 39	0.627
As part of TEI (%)	19 ± 5	19 ± 5	0.867
Fat intake			
Total amount (g/day)	54 ± 26	48 ± 23	0.153
As part of TEI (%)	27 ± 6	25 ± 6	0.033 *
Saturated fat intake (g/day)	13 ± 7	11 ± 6	0.043 *
Fiber intake (g/day)	6 ± 7	6 ± 7	0.960
Sugar intake (g/day)	35 ± 28	36 ± 28	0.833
Food group intake			
Cereals and cereal products (g/day)	401 ± 190	399 ± 188	0.964
Fast food (g/day)	46 ± 5	56 ± 8	0.294
Meat and poultry (g/day)	156 ± 14	139 ± 21	0.489
Fish and seafood (g/day)	68 ± 8	71 ± 10	0.776
Eggs (g/day)	34 ± 33	25 ± 3	0.034 *
Legumes (g/day)	14 ± 4	15 ± 3	0.817
Milk and dairy products (g/day)	112 ± 14	65 ± 12	0.012 *
Vegetables (g/day)	36 ± 3	50 ± 7	0.076
Fruits (g/day)	217 ± 24	238 ± 44	0.641
Beverages (g/day)	353 ± 36	345 ± 51	0.895
Confectionaries (g/day)	42 ± 6	45 ± 5	0.772
Bread spreads (g/day)	3 ± 0.4	3 ± 1	0.983
Condiments (g/day)	20 ± 2	23 ± 3	0.442

AGT: abnormal glucose tolerance; NGT: normal glucose tolerance; TEI: total energy intake. Continuous data expressed in mean ± SD/SE, analysed using independent *t*-test. * *p* < 0.05. Food groups are based on the original food groups in Malaysian Adult Nutrition Survey 2014 food frequency questionnaire (Institute for Public Health, 2014).

**Table 3 nutrients-15-02819-t003:** Correlations of dietary patterns with energy and macronutrient intakes (*n* = 150).

Energy and Macronutrient Intakes	DP 1 (Unhealthy)	DP 2 (Fish-Eggs-Fruits-Vegetables)	DP 3 (Cereals-Confectionaries)	DP 4 (Legumes-Dairy)	DP 5 (Meat-Sugar-Sweetened Beverages)
TEI (kcal/day)	0.413 ***	0.450 ***	0.466 ***	0.240 **	0.424 ***
Carbohydrate (g/day)	0.438 ***	0.304 ***	**0.634** ***	0.270 **	0.225 *
Protein (g/day)	0.189 *	**0.571** ***	0.188 *	0.111	**0.545** ***
Total fat (g/day)	0.414 ***	0.420 ***	0.191 *	0.221 **	**0.528** ***
Saturated fat (g/day)	0.414 ***	0.354 ***	0.035	0.273 **	**0.610** ***
Fiber (g/day)	0.136	**0.514** ***	0.237 **	**0.373** ***	0.089
Sugar (g/day)	0.379 ***	0.409 ***	0.297 ***	**0.355** ***	0.099

DP: dietary pattern. Bold indicates moderate correlation (Pearson’s R > 0.5). * *p* < 0.05, ** *p* < 0.01, *** *p* < 0.001.

**Table 4 nutrients-15-02819-t004:** Multivariate association of dietary patterns with abnormal glucose tolerance (*n* = 150).

Dietary Pattern	Model 1		Model 2		Model 3	
	OR	95% CI	OR	95% CI	OR	95% CI
DP 1 (Mostly unhealthy)	1.060	0.761, 1.476	1.035	0.723, 1.481	1.054	0.697, 1.595
DP 2 (Fish-eggs-fruits-vegetables)	0.886	0.624, 1.260	0.879	0.604, 1.278	0.857	0.565, 1.300
DP 3 (Cereals-confectionaries)	1.281	0.912, 1.800	1.406	0.971, 2.036	1.536 *	1.002, 2.354
DP 4 (Legumes-dairy)	0.937	0.657, 1.336	0.985	0.694, 1.398	0.986	0.692, 1.406
DP 5 (Meat-sugar-sweetened beverages)	0.812	0.571, 1.155	0.825	0.565, 1.205	0.794	0.522, 1.206

Model 1: unadjusted. Model 2: adjusted for age, education, occupation, household size, family history of diabetes. Model 3: adjusted for all variables in model 2 and total energy intake. CI: confidence interval; DP: dietary pattern; GDM: gestational diabetes mellitus; OR: odds ratio. * *p* < 0.05.

## Data Availability

Not applicable.

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
