# Peer review of "Dietary Patterns Associated with Abnormal Glucose Tolerance following Gestational Diabetes Mellitus: The MyNutritype Study"

_nutrients, 2023, doi:10.3390/nu15122819_

Round 1

Reviewer 1 Report

This study evaluated 157 Asian women post GDM, enrolled in the MyNutritype cohort. The association of 5 different dietary patterns with AGT was evaluated cross-sectionally, on the ‘a posteriori diet’. The topic is important, because AGT is becoming a global burden. Only one pattern (the Cereals-confectionaries dietary pattern) was significantly associated with AGT.

Introduction is too concise and can be expanded, including more information.

Material and Methods: I found that the categorization of the different patterns (Five dietary patterns were derived using principal component analysis: ‘Unhealthy’, ‘Fish-eggs-fruits-vegetables’, ‘Cereals-confectionaries’, ‘Legumes-dairy’, and 36 ‘Meat-sugar-sweetened-beverages’) is quite odd: authors stated that they have been identified according principal component analysis, but authors should question whether they have any practical use for the results and how they can be compared with the standard patterns used in literature.

Results and discussion: the above criticism influence their contents.

In conclusion, I believe that the article should be much improved: this kind of dietary categorization is not useful to draw information and associations, since it’s already well known that sugars and refined carbohydrates negatively influence glucose metabolism.

How about the effect (protective and harmful) of the different food groups? The sample is relatively small, and a more in-deep analysis should be performed.

Author Response

Response to Reviewer 1 Comments

Comment 1:

This study evaluated 157 Asian women post GDM, enrolled in the MyNutritype cohort. The association of 5 different dietary patterns with AGT was evaluated cross-sectionally, on the ‘a posteriori diet’. The topic is important, because AGT is becoming a global burden. Only one pattern (the Cereals-confectionaries dietary pattern) was significantly associated with AGT.

Response 1:

I would like to thank the reviewer for his/her insightful comments which absolutely helped in improving the quality and readability of this manuscript.

Comment 2:

Introduction is too concise and can be expanded, including more information.

Response 2:

We agree with this comment and have added further information that may be useful to the readers. First of all, PCA was chosen for this study as it is the most common method to analyze dietary patterns with proven reproducibility (Zhao et al., 2021). We have added the following sentences to Introduction (Page 2, paragraph 3-4, lines 70 – 86):

“The more prominent method to assess dietary patterns in women post-GDM appeared to be a priori, which assessed the adherence to dietary quality indices or healthy dietary guidelines including the alternate Mediterranean diet (aMED), Dietary Approaches to Stop Hypertension (DASH), or alternate Healthy Eating Index (aHEI) [19,20]. Nonetheless, both of these studies used data from the Nurses’ Health Study II which involved mostly Caucasian women [19,20], whose dietary quality, nutrient intakes and food sources had been reported to be different compared to Asian women [21, 22]. While a priori indices evaluate the healthiness of a diet based on existing dietary guidelines, they are restricted by existing knowledge of diet-disease relationships [17].

On the other hand, a posteriori approaches, such as principal component analysis (PCA), help to characterize the existing dietary habits in a population. PCA evaluates the overall habitual diet and the diversity in dietary intake of a population [23]. During analysis, the uncorrelated patterns represent the various dietary characteristics of the population and may be used to predict health outcomes; hence, the results may be more meaningful and significant compared to single-nutrient or single-food dietary analyses [23]. Furthermore, several major dietary patterns derived using this method had been replicated across different populations [23].”

We have expanded on Malaysia being a multi-ethnic country with diverse gastronomic cultures, which made PCA as the chosen method that can meaningfully analyze the diverse dietary patterns of this country (Page 2, paragraph 5, 89 -101):

“Malaysia is also in the midst of the Asian epidemic of T2D [24]. Malaysia is composed of diverse ethnic groups, primarily Malays, Chinese, and Indians [24]. The Chinese and Indian communities Malaysia were descendants from their ancestors in China and India, respectively, who migrated to Malaysia between early 19th to mid-20th century [25]. If India and China were projected to exceed the world’s global estimates of T2D, their respective counterparts in Malaysia would be worse [24]. The interactions between different ethnic groups in Malaysia introduced varying degrees of genetic combinations and cultural influences [26] which possibly influenced the diet as well. The epidemiological transition brought upon by urbanization has also influenced Malaysia into undergoing rapid nutrition transition, indicated by the modernization and Westernization of traditional diets [27]. Hence, using the PCA to analyze dietary patterns may be more helpful in depicting the diverse dietary habits and planning individualized dietary strategies, specifically in multi-ethnic populations including in Malaysia.”

Comment 3:

Material and Methods: I found that the categorization of the different patterns (Five dietary patterns were derived using principal component analysis: ‘Unhealthy’, ‘Fish-eggs-fruits-vegetables’, ‘Cereals-confectionaries’, ‘Legumes-dairy’, and 36 ‘Meat-sugar-sweetened-beverages’) is quite odd: authors stated that they have been identified according principal component analysis, but authors should question whether they have any practical use for the results and how they can be compared with the standard patterns used in literature.

Results and discussion: the above criticism influence their contents.

Response 3:

We would like to thank the reviewer for pointing this out. We have amended the manuscript to better indicate the practical use of the results.

We have compared our PCA results with PCA-derived dietary patterns from previous studies. We have discussed the dietary patterns in the context of what is known about dietary patterns in Malaysia and its neighbouring country, Singapore, that has similar cultural diversity (Page 13, paragraph 2, lines 389 – 403):

“In a multi-ethnic, diverse population undergoing rapid nutrition transition like Malaysia, the effort is specifically useful to identify existing variations in dietary intakes. Unlike one or two major dietary patterns that are reported in Western countries (such as ‘Healthy/Prudent’ and ‘Unhealthy’ dietary patterns), the diversity in Asian countries, including Malaysia, informs on the practical use of culturally tailored dietary patterns as the ‘One size fits all’ approach may not be suitable. The PCA-derived dietary patterns in this study did not mimic the usual dietary patterns reported in Western countries due to differences in dietary habits. For instance, the ‘Sugar-sweetened-beverages (SSB)’ dietary pattern was associated with increased cardiometabolic risks in Malaysian multi-ethnic healthy adults [51]. Additionally, a dietary pattern characterized by higher intake of dim sum, meat/processed meat, SSB and sweetened foods, and fried foods were associated with higher risk of T2D in Singaporean Chinese adults [62]. These results were almost similar with our findings on women post-GDM, and informed the need for more tailored dietary intervention strategies to facilitate disease prevention. Dietary patterns identified in this study indicated the most common foods in diet of women post-GDM.”

Comment 4:

In conclusion, I believe that the article should be much improved: this kind of dietary categorization is not useful to draw information and associations, since it’s already well known that sugars and refined carbohydrates negatively influence glucose metabolism. How about the effect (protective and harmful) of the different food groups? The sample is relatively small, and a more in-deep analysis should be performed.

Response 4:

Although it’s known that sugars and refined carbohydrates negatively influence glucose metabolism, the composition of the dietary pattern characterizing sugars/carbohydrates is varying based on the local diet, which may provide more useful information when tailoring dietary intervention strategies to prevent AGT in women post-GDM.

We have elaborated the dietary patterns in more detail, including the protective and harmful effects of the different food groups (Results, Page 8-9, lines 259 - 311). We have also added in a figure to better illustrate the food groups (Figure 1). We believe that through a more in-depth description of the dietary patterns we are able to showcase the potential for individualisation of dietary recommendations.

“The first dietary pattern, ‘Unhealthy’, was characterized by high intakes of fast food; milk and dairy products; coffee, tea, and sugar-sweetened beverages (SSB); sugar, honey, bread spreads, and creamer; and condiments. This dietary pattern contributed to 19.9% of the total variance. This dietary pattern was labelled ‘Unhealthy’ because most of the foods that loaded highly in this pattern were demonstrated to exert detrimental effects on metabolic health. Although total dairy, low-fat dairy, cheese, and yogurt intakes had been shown to have protective effects against T2D [49], this food group also included both full-fat and low-fat milk/dairy, and fresh milk and flavoured milk (e.g. chocolate milk). Flavoured milk is often sweetened; one cup of chocolate milk contains approximately 16g of total sugars, compared to full-fat milk which has 11g of total sugars [35]. Fast food has been associated with weight gain and insulin resistance, which could lead to an increased risk of obesity and T2D [50]. Coffee and tea were mostly consumed with condensed milk or sugar among Malaysian adults, and SSB had been associated with increased cardiometabolic risks in the Malaysian population [51]. Sugar, honey, bread spreads, and creamer (this food group also contains condensed milk) are high in added sugar, which had been shown to increase cardiovascular risk by increasing LDL-cholesterol and triglyceride levels [52]. Lastly, condiments, which includes table salt, a variety of sauces (soy, chili, ketchup, oyster, fish sauces), and dipping sauces (sambal, budu, cincalok) are high in sodium content, which increased blood pressure levels [53].

The second dietary pattern derived, ‘Fish-eggs-fruits-vegetables’, included high intakes of fish and seafood; eggs; and fruits and vegetables; and explained 12.5% of the total variance. Marine fish and green leafy vegetables are among the top ten foods most consumed by Malaysian adults [54]. Aside from eggs, the foods in this dietary pattern was included in the ‘Mainly healthy’ dietary pattern which was found to be inversely correlated with the risk of T2D [55].  While this dietary pattern among our subjects included several healthy food groups and was correlated with higher fiber intake, it also showed the highest correlation among the groups with sugar intake (Table 3). The correlation with energy intake also suggests that attention to portion control may sometimes be required in those adhering to this pattern.

The third dietary pattern, ‘Cereals-confectionaries’ consisted of high intakes of cereals and grains and confectionaries, but low intake of milk and dairy products (accounting for 9.5% of the total variance). The cereals included a variety of grains, cereals, and cereal products, including rice (white rice, brown rice, flavoured rice), noodles, pasta, corn, breakfast cereals, oats, breads, and local flatbread. Confectionaries consisted of sweet or savoury foods and snacks, such as local kuih, biscuits, chocolate, candies, and pastries.

Next, the ‘Legumes-dairy’ dietary pattern had high intakes of legumes, and milk and dairy products (explained 9.2% of the total variance). Total legume intakes were associated with an inverse risk of T2D [56], as well as dairy intake [49]. Like in dietary pattern 1 (‘Unhealthy’), however, the milk and dairy products in this dietary pattern included both low-fat and full-fat milk/dairy, and fresh milk and flavoured milk. Thus, this dietary pattern could have a mixed effect on the risk of AGT.

The fifth dietary pattern, ‘Meat-sugar-sweetened beverages’ which accounted for 8.7% of the total variance, was characterized by high intakes of meat and poultry and SSB. The meat in this dietary pattern referred to meat and poultry (including beef, mutton, chicken, duck, pork) and their products (such as internal organs, ham, bacon, luncheon meat, meatballs). Red and processed meat had been associated with a positive risk of T2D, whereas poultry with decreased risk of T2D [55]. SSB included coffee, tea, which are usually added with sugars among Malaysians [51], as well as non-alcoholic sugary drinks such as malted drinks, yogurt drinks, carbonated drinks, and fruit juices (Figure 1).”

Thus, each dietary pattern identified potential targets to improve the healthfulness of the diet, facilitating individualisation of recommendations in clinical practice.

Figure 1. Factor loading scores of dietary patterns identified using principal component analysis. [Please see the attachment for figure]

DP 1— ‘Unhealthy’ dietary pattern; DP 2— ‘Fish-eggs-fruits-vegetables’ dietary pattern; DP 3— ‘Cereals-confectionaries’ dietary pattern; DP 4— ‘Legumes-dairy’ dietary pattern; DP 5— ‘Meat-sugar-sweetened beverages’ dietary pattern

Red circle    : food group perceived to be healthy; blue circle    : food group perceived to be unhealthy; purple circle   : food group consisting of healthy and unhealthy food items

We hope that our answers are adequate to address your comments. Thank you.

Reviewer 2 Report

This is good , but limited research. Limitations are in its structure- retrospective a posteriori-  investigation on dietary habits. Furhter limitation - Malaysia. Statistics- PCA 59% variance. Would like to see 2D pic presenting grouping instead of numerous tables. Authors also state that bias of the researchers is also problem, as well as the heterogeneity of population.    So it might be interesting for the readers to see results of Malasian study. 

Author Response

Response to Reviewer 2 Comments

Comment: This is good, but limited research. Limitations are in its structure- retrospective a posteriori- investigation on dietary habits. Further limitation - Malaysia.

Statistics- PCA 59% variance.  Would like to see 2D pic presenting grouping instead of numerous tables. Authors also state that bias of the researchers is also problem, as well as the heterogeneity of population.  So it might be interesting for the readers to see results of Malaysian study. 

Response: Thank you for the feedback. We have noted the limitations of PCA to analyze a posteriori dietary patterns on Page 14, paragraph 3, lines 438 - 442. This includes possible researcher bias, such as during categorization or labelling of the food groups, and deciding the number of factors to retain. We have clarified further with the following sentence (Page 14, paragraph 3, lines 442 – 448):

“The subjective decision making during PCA, and the heterogeneity of the population, has made comparisons with other studies difficult. However, the study maintained primary considerations when conducting PCA, such as the nutrient composition of the food [72], the key foods constituting dietary patterns that are relevant in preventing AGT [72], and the rapidly shifting dietary landscape in Malaysia (including the Westernization and modernization of traditional diets; and the availability of convenience or processed foods) [27,38].”

We also acknowledge the limitation of the setting of the study. We added the sentences (Page 14, paragraph 2, lines 435 – 437):

“The study was conducted in Malaysia, and food intake was assessed using a culturally specific FFQ that is unique to this country. Thus, the results may not be generalizable to other countries.”

Regarding the statistics, we added the sentence (Page 14, paragraph 3, lines 448 – 450):

“Nonetheless, the dietary patterns derived using the PCA approach here explained 59% of the population variance. This compares more favourably to previous explorations in Malaysia [27,38].”

We have removed Table 3 and put it under Supplementary (Table S1). We substituted it with a 2D picture (Radar chart) presenting the food grouping on Page 10:

Figure 1. Factor loading scores of dietary patterns identified using principal component analysis. [Please see the attachment for figure]

DP 1— ‘Unhealthy’ dietary pattern; DP 2— ‘Fish-eggs-fruits-vegetables’ dietary pattern; DP 3— ‘Cereals-confectionaries’ dietary pattern; DP 4— ‘Legumes-dairy’ dietary pattern; DP 5— ‘Meat-sugar-sweetened beverages’ dietary pattern

Red circle    : food group perceived to be healthy; blue circle   : food group perceived to be unhealthy; purple circle   : food group consisting of healthy and unhealthy food items

We hope that our answers are adequate to address your comments. Thank you.

Round 2

Reviewer 1 Report

Please check in the figure if the legenda regarding the color of the points